# Anion-capped metallohost allows extremely slow guest uptake and on-demand acceleration of guest exchange

Yoko Sakata[1], Chiho Murata[1] & Shigehisa Akine[1]

The switching of molecular recognition selectivity is important for tuning molecular functions based on host–guest binding. While the switching processes in artificial functional molecules are usually driven by changes of the thermodynamic stabilities, non-equilibrium phenomena also play an important role in biological systems. Thus, here we designed a host–guest system utilizing a non-equilibrium kinetically trapped state for on-demand and time-programmable control of molecular functions. We synthesized a bis(saloph) macrocyclic cobalt(III) metallohost $1(OTf)_2$, which has anion caps at both sides of the cation-binding site. The anion caps effectively retard the guest uptake/release so that we can easily make a non-equilibrium kinetically trapped state. Indeed, we can obtain a long-lived kinetically trapped state $\{[1\bullet K]^{3+} + La^{3+}\}$ prior to the formation of the thermodynamically more stable state $\{[1\bullet La]^{5+} + K^+\}$. The guest exchange to the more stable state from this kinetically trapped state is significantly accelerated by exchange of $TfO^-$ anion caps by $AcO^-$ in an on-demand manner.

[1] Graduate School of Natural Science and Technology, Kanazawa University, Kakuma-machi, Kanazawa 920-1192, Japan. Correspondence and requests for materials should be addressed to S.A. (email: akine@se.kanazawa-u.ac.jp).

The switching of molecular recognition selectivity is important for tuning molecular functions based on host–guest binding[1–5]. Most of these switching processes are driven by a stimuli-responsive guest uptake/release or guest exchange upon the structural change between two different states. To date, a number of responsive molecular recognition systems has been developed by taking advantage of various kinds of stimuli such as redox reactions[6–8], acid/base reactions[9–11], light irradiation[12–15] and metal ion addition[16,17], which lead to drastic structural changes in the host scaffold. Such stimuli-responsive systems can also be found in nature, as represented by $Na^+/K^+$-ATPase, which pumps $Na^+$ out of a cell with the concomitant transport of $K^+$ into the cell against their concentration gradients with the aid of ATP[18,19]. The key for regulating the concentrations is the structural changes in the enzyme between the two main states, E1 and E2, which selectively bind $Na^+$ and $K^+$, respectively. In general, the selectivity switching requires reversal of the thermodynamic stabilities, which can be evaluated by the association constants, $K_a$, in the two different states (Fig. 1a).

In addition, non-equilibrium phenomena play an important role in biological systems as well as in supramolecular polymer systems[20–30]. For example, passive transport finely controls ion uptake and release using ion channels and carriers according to their concentration gradients. We could make similar ion uptake/release systems using much simpler artificial host molecules if we can access a suitable non-equilibrium kinetically trapped state (Fig. 1b(i)). Once a kinetically trapped state is generated, removal of the kinetic barrier would initiate the uptake/release of guest ions according to the intrinsic affinity difference of the host molecule (Fig. 1b(ii)). This would enable us to tune the molecular functions in an on-demand and time-programmable manner as seen in biological systems. However, such a guest uptake/release based on a kinetically trapped state has rarely been utilized for controlling molecular functions in artificial systems. As to generating a kinetically trapped state as the first step (Fig. 1b(i)), there have been several host molecules that temporarily bind a kinetically favoured guest prior to the binding with a thermodynamically favoured guest[31–43]. Therefore, we envisioned a kinetically trapped host–guest system having the second step in which removal of the kinetic barrier can initiate the guest uptake or exchange (Fig. 1b(ii)).

We designed a host molecule $[LCo_2(CH_3NH_2)_4](OTf)_2$ ($= 1(OTf)_2$) containing an 18-crown-6-like cavity suitable for cation recognition[44,45]. The molecule has two octahedral cobalt(III) centres and four axially coordinating methylamine molecules as anchors, which would contribute to holding anionic species at the capping sites (Fig. 1c). The anion caps are expected to block the guest from entering and exiting the cavity, and work as a removable kinetic barrier (Fig. 1d). In this paper, we report an ion recognition system that shows an extremely slow uptake of guest cations due to the blocking effect of the anion caps. This enables us to obtain a kinetically trapped state, from which on-demand acceleration of the guest exchange is triggered by the anion cap exchange.

## Results

### Synthesis of anion-capped metallohost.
The metallohost $1(OTf)_2$ was synthesized by the complexation of the bis(saloph) macrocyclic host[44,45] ($H_2$saloph $= N,N'$-disalicylidene-$o$-phenylenediamine) with cobalt(II) acetate under aerobic conditions in the presence of methylamine (Supplementary Fig. 1). Each of the two saloph moieties accommodates a low-spin diamagnetic cobalt(III) ion, to which two methylamine molecules coordinate to form an octahedral geometry. A crystallographic analysis revealed its molecular structure in which the

central $O_6$ cavity remains vacant as similarly observed in the nickel(II) analogue[44,45]. The four methylamine molecules coordinate to the octahedral cobalt(III) centres to form the capping sites as expected, but the triflate counteranions were not located at the capping sites (Supplementary Fig. 15).

### Guest recognition behaviour.
We were initially skeptical about the idea that the dicationic metallohost $1^{2+}$ would bind cationic guests because electrostatic repulsion between the positive charges was expected. However, $1^{2+}$ showed an excellent binding affinity towards cationic guests. For example, the addition of 1 equiv of NaOTf to the metallohost $1(OTf)_2$ in $CD_3OD$ resulted in complete conversion to the inclusion complex $[1\bullet Na]^{3+}$ (Fig. 2a and Supplementary Fig. 2). It is noteworthy that the $^1H$ NMR signals for the inclusion complex were separately observed from those for the free metallohost $1^{2+}$. This indicated that the complexation/decomplexation process is slow on the NMR timescale. This slow guest exchange would be due to the anion caps as we had expected. Indeed, the X-ray crystallographic analysis of the $Na^+$ complex (Fig. 2b) clearly showed that the counteranions are located at the capping sites. The $Na^+$ ion was located in the central $O_6$ site, and the two triflate anions at the capping sites coordinate to the guest $Na^+$ ion from both sides of the macrocycle. In addition, the anions are hydrogen-bonded to the methylamine molecules as the anchors. The strong interaction between these triflate counteranions and guest $Na^+$ ion is probably one of the important factors for the unexpectedly strong cation-binding affinity to overcome the electrostatic repulsion between the positive charges.

### Thermodynamics and kinetics of cation recognition.
While the metallohost $1(OTf)_2$ did not interact with the smaller ions ($Mg^{2+}$, $Li^+$), it strongly recognized metal ions having radii of $> 1.26$ Å (Supplementary Figs 3–11). In particular, the association constants for metal ions having ionic radii of $\sim 1.3$ Å ($Ca^{2+}$, $La^{3+}$, $Na^+$ and $K^+$) were greater than $10^6 M^{-1}$ (Table 1, equations (1) and (2)).

$$1^{2+} + M^{n+} \underset{k_{out}}{\overset{k_{in}}{\rightleftarrows}} [1\bullet M]^{(2+n)+} \qquad (1)$$

$$K_a = \frac{\left[[1\bullet M]^{(2+n)+}\right]}{[1^{2+}][M^{n+}]} = \frac{k_{in}}{k_{out}} \qquad (2)$$

The binding strengths seem to be mainly governed by the size-fit principle rather than the electrostatic interactions, although the binding strengths might be influenced by some other factors such as the charge of the guest ions. Even multivalent metal ions, such as $Ca^{2+}$ or $La^{3+}$, were strongly bound to the dicationic metallohost $1(OTf)_2$ in spite of the expected electrostatic repulsion. This unexpectedly strong binding can be ascribed to the interaction with the anion caps as observed for the $Na^+$ complex (for the X-ray crystal structures of the $K^+$ and $Ca^{2+}$ complexes, see Supplementary Figs 17 and 18).

As expected based on the structures of the host–guest complexes having anion caps, the guest uptake and release were extremely slow. For example, the uptake of better guests, such as $K^+$, $Na^+$ and $Ca^{2+}$, was slow on the NMR timescale (Table 1). Nevertheless, the guest uptake was completed and reached equilibrium within 5 min after mixing the guests with $1(OTf)_2$. In stark contrast, the complexation with $La(OTf)_3$ was exceptionally slow (Supplementary Fig. 13). It took 120 h for complete encapsulation after the addition of 2 equiv of $La(OTf)_3$. The half-life of the guest uptake was roughly estimated to be $\sim 10$ h from the time course analysis of the $^1H$ NMR spectra. Thus, the

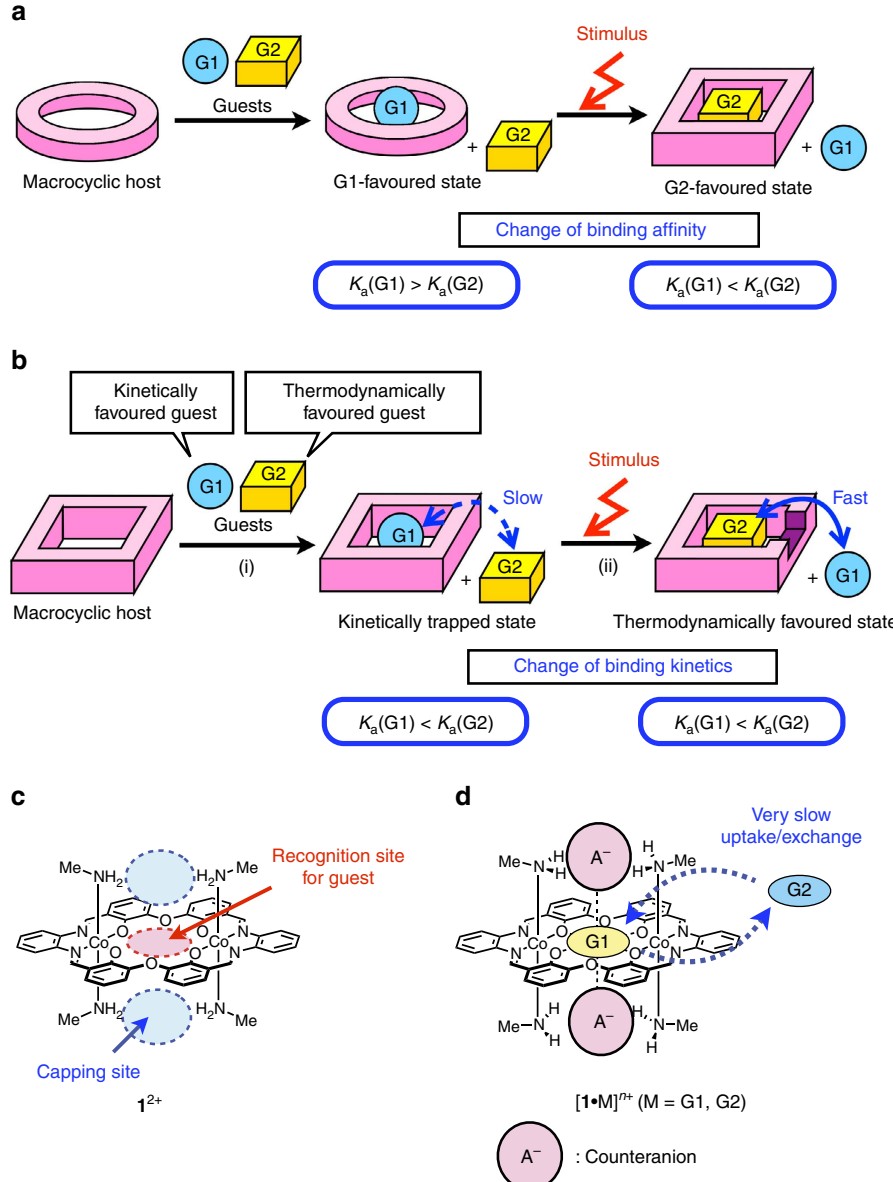

**Figure 1 | Design and concept for kinetically controlled on-demand acceleration of guest uptake/exchange. (a)** Schematic illustration of thermodynamically controlled stimuli-responsive guest exchange in which the association constants for two different of guests are reversed by an external stimulus. (**b**) Schematic illustration of kinetically controlled stimuli-responsive guest exchange in which the kinetic barrier for the guest exchange is removed by an external stimulus. (**c**) Design of macrocyclic host $1^{2+}$ with two capping sites. (**d**) Concept of the extremely slow guest uptake/exchange by taking advantage of anion caps.

guest inclusion by $1(OTf)_2$ was much slower than that by simple crown ethers (within milliseconds to seconds)[46–48]. The retarded guest inclusion should mainly be attributed to the effect of the anion caps, although there are differences in the charge and the host structures that could also affect the guest uptake rates.

**On-demand acceleration of guest exchange.** We expected that this exceptionally slow uptake of $La^{3+}$ is due to the capping effect of the triflate anions. We thus expected that the guest uptake rates can be tuned by changing the counteranions because the capping effect might depend on the sizes and basicity of the anions. It was inferred that it would be easy to understand the effect of counteranions if we could start with the metallohost having non-coordinating anion such as $BPh_4^-$. However, it was difficult

to obtain pure sample of $1(BPh_4)_2$. Thus, 3 equiv of $Bu_4NX$ ($X = AcO^-$, $F^-$, $Cl^-$, $Br^-$, $BF_4^-$, $PF_6^-$, $BPh_4^-$) was added to the mixture of $1(OTf)_2$ and $La(OTf)_3$ and the uptake rates were compared with the blank experiment. As we expected, acetate ion ($AcO^-$) significantly accelerated the cation uptake, while other anions ($F^-$, $Cl^-$, $Br^-$, $BF_4^-$, $PF_6^-$, $BPh_4^-$) did not. This acceleration was also observed when $La(OAc)_3$ was used as the $La^{3+}$ source. The uptake of $La^{3+}$ was completed within 5 min after the addition of 2 equiv of $La(OAc)_3$ (Supplementary Figs 12 and 14). The uptake rate $k_{in}$ of $La(OAc)_3$ was estimated to be $\geq 30\,M^{-1}$ $s^{-1}$, which was at least 100 times faster than that of $La(OTf)_3$. Clearly, the counteranions played a critical role in the exceptionally slow uptake of $La(OTf)_3$.

As already described, the guest uptake/release by $1^{2+}$ is exceptionally slow, and this would allow us to obtain kinetically

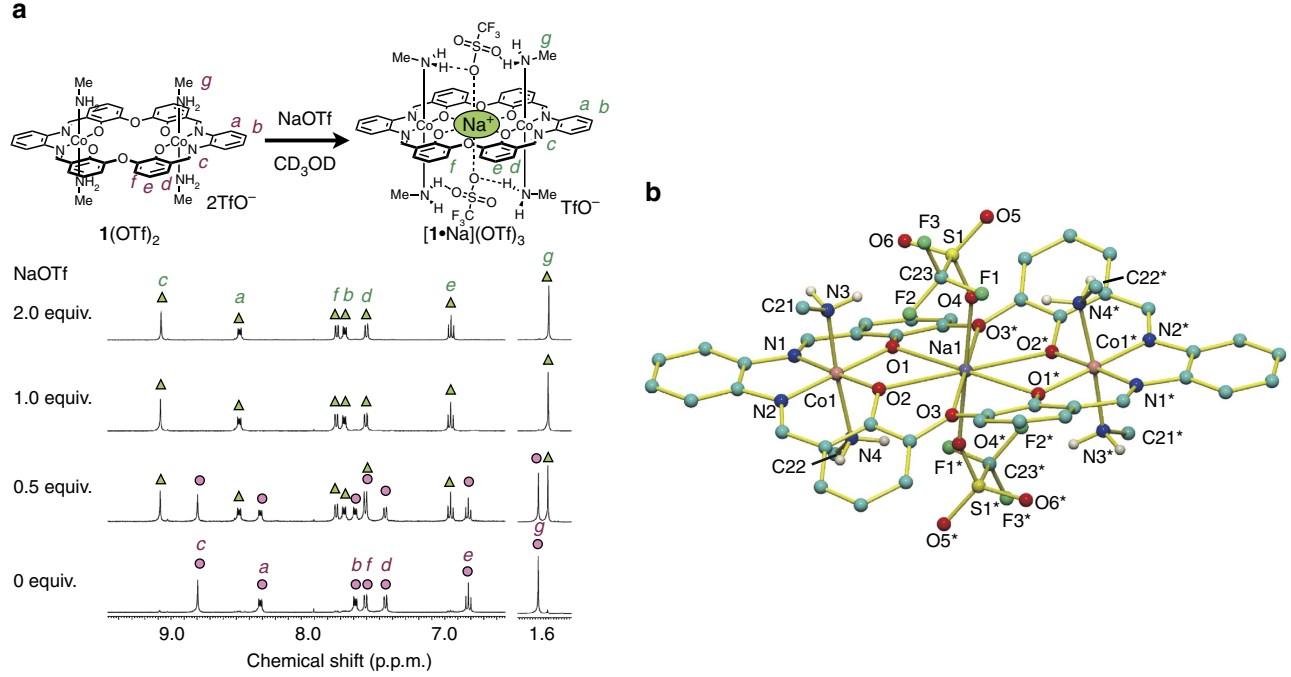

**Figure 2 | Na$^+$ encapsulation by anion-capped metallohost.** (**a**) $^1$H NMR spectral changes of **1**(OTf)$_2$ upon the addition of NaOTf in CD$_3$OD (400 MHz, [**1**(OTf)$_2$] = 1.0 mM). (**b**) X-ray crystal structure of [**1**•Na(OTf)$_2$](OTf). Hydrogen atoms are omitted for clarity except for those of the NH$_2$ groups. The solvent molecules and one triflate anion are also omitted.

**Table 1 | Association constants and kinetic data for the complexation between 1$^{2+}$ or 18-crown-6 and guest cations.**

| Guest* | Ionic radius (Å)[52] | Association constant $K_a$ (M$^{-1}$) | Kinetic data | | Association constant of 18-crown-6 $K_a$ (M$^{-1}$) |
|---|---|---|---|---|---|
| | | | $k_{in}$ (M$^{-1}$s$^{-1}$) | $k_{out}$ (s$^{-1}$) | |
| Mg$^{2+}$ | 1.03 | n.d.$^\dagger$ | n.d.$^\dagger$ | n.d.$^\dagger$ | $4.1 \times 10^3$ (ref. 53) |
| Li$^+$ | 1.06 | n.d.$^\dagger$ | n.d.$^\dagger$ | n.d.$^\dagger$ | $\sim$0 (ref. 54) |
| Ca$^{2+}$ | 1.26 | $9.5 \times 10^6$$^\ddagger$ | $\geq 30$ | $<1$$^\P$ | $9.1 \times 10^3$ (ref. 53) |
| La$^{3+}$ | 1.30 | $2.4 \times 10^6$$^\ddagger$ | $\sim 10^{-2\|}$ | $\sim 10^{-8}$ | $5.5 \times 10^3$ (ref. 53) |
| Na$^+$ | 1.32 | $8.5 \times 10^6$$^\ddagger$ | $\geq 30$ | $<1$$^\P$ | $2.9 \times 10^4$ (ref. 53) |
| K$^+$ | 1.65 | $1.1 \times 10^6$$^\ddagger$ | $1.9 \times 10^7$ | $17$$^\P$ | $1.6 \times 10^6$ (ref. 53) |
| Rb$^+$ | 1.75 | $\sim 2 \times 10^5$$^\S$ | $\sim 3 \times 10^7$ | $150$$^\P$ | $5.4 \times 10^5$ (ref. 53) |
| Cs$^+$ | 1.88 | $2.3 \times 10^3$$^\S$ | $6.9 \times 10^6$ | $3,000$$^\P$ | $3.1 \times 10^4$ (ref. 53) |

n.d., not determined.
*Triflate salt.
$^\dagger$Not determined due to the weak interaction.
$^\ddagger$Determined by competitive experiments using 18-crown-6.
$^\S$Obtained by a nonlinear least-square analysis of the spectral change.
$^\|$Obtained by the time course analysis of the $^1$H NMR spectra during the encapsulation of La(OTf)$_3$.
$^\P$Determined by the line-shape analysis of the imine signals of the $^1$H NMR spectra, where 50% of each metal ion is encapsulated in the metallohost.

trapped states as depicted in Fig. 1b(i). In order to use this system for kinetically controlled switching, it is necessary to accelerate it by removing or lowering the kinetic barrier (Fig. 1b(ii)). Since the counteranions (TfO$^-$ and AcO$^-$) significantly affect the guest uptake/release rates of $1^{2+}$, we expected that exchanges of the anion caps would lower the barrier and trigger the guest uptake/release. Thus, we made an on-demand guest exchange system driven by the anion cap replacement.

We chose K$^+$ and La$^{3+}$ as the guests to obtain a kinetically trapped state because La(OTf)$_3$ was more strongly bound ($K_a = 2.4 \times 10^6$ M$^{-1}$) than KOTf ($K_a = 1.1 \times 10^6$ M$^{-1}$), but more slowly taken up ($k_{in} \approx 10^{-2}$ M$^{-1}$s$^{-1}$) than KOTf ($k_{in} = 1.9 \times 10^7$ M$^{-1}$s$^{-1}$). As expected from the thermodynamic and kinetic data, we successfully generated the kinetically trapped state {[$1$•K]$^{3+}$ + La$^{3+}$} when KOTf (1 equiv) and La(OTf)$_3$

(2 equiv) were added to the metallohost $1$(OTf)$_2$ at once (Figs 3a(i) and 4(i)). The kinetically formed K$^+$ complex would be converted to the thermodynamically more stable La$^{3+}$ complex according to the difference in the association constants. However, this guest exchange was found to be very slow; the K$^+$ complex was still dominant even after 14 days (Figs 3a(i),b(i) and 4(ii) and Supplementary Fig. 20). It should be noted that the reverse reaction from [$1$•La]$^{5+}$ to [$1$•K]$^{3+}$ did not take place at all when KOTf was added to the La$^{3+}$ complex (Fig. 4(iii) and Supplementary Fig. 23). Since both of the two states, {[$1$•K]$^{3+}$ + La$^{3+}$} and {[$1$•La]$^{5+}$ + K$^+$}, were observable, the thermodynamically less stable state, {[$1$•K]$^{3+}$ + La$^{3+}$}, is thought to be a very stable kinetically trapped state.

Interestingly, the addition of acetate ion markedly changed this situation. When 3 equiv of Bu$_4$NOAc was present from the

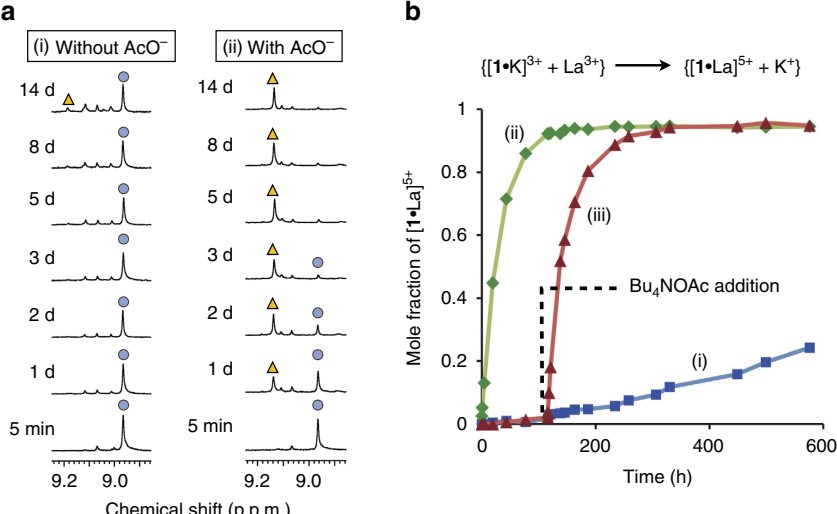

**Figure 3 | Time course analysis of guest exchange from [1•K]$^{3+}$ to [1•La]$^{5+}$.** (a) Changes of the imine signals in the $^1$H NMR spectra (blue circle: K$^+$ complex and yellow triangle: La$^{3+}$ complex) in the absence/presence of tetrabutylammonium acetate (1.0 mM, CD$_3$OD, 25 °C). (b) Plots of mole fractions of [1•La]$^{5+}$ versus time after addition of K$^+$ and La$^{3+}$.

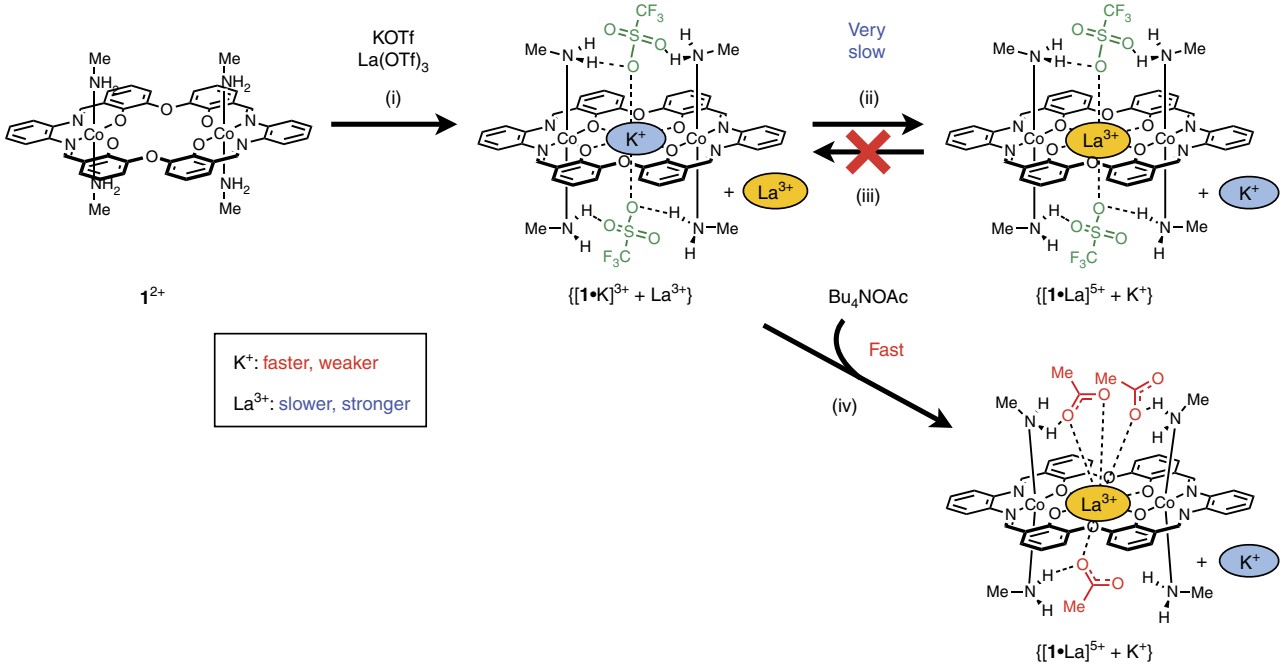

**Figure 4 | Guest recognition and exchange behaviours of 1(OTf)$_2$.** When KOTf and La(OTf)$_3$ were added to the metallohost 1(OTf)$_2$ at once, the kinetically trapped state {[1•K]$^{3+}$ + La$^{3+}$} was generated (i). While the conversion from the kinetically formed K$^+$ complex to the thermodynamically more stable La$^{3+}$ complex was exceptionally slow (ii, iii), the guest exchange was significantly accelerated by the addition of tetrabutylammonium acetate (iv).

beginning, the exchange of K$^+$ with La$^{3+}$ took place much faster (Figs 3a(ii),b(ii) and 4(iv) and Supplementary Fig. 21). The reaction was accelerated by $\sim$75 times, which was estimated from the initial rates. Obviously, the addition of Bu$_4$NOAc drastically accelerated the guest exchange. It is noteworthy that we can accelerate this metal exchange at any stage of the reaction. For example, when 3 equiv of Bu$_4$NOAc was added after 120 h, [1•La]$^{5+}$ started to immediately increase after the addition (Fig. 3b(iii) and Supplementary Fig. 22). Since almost no metal exchange had occurred before the addition, the acetate ion acted as a trigger to initiate the guest exchange reaction from the kinetically trapped state.

In the X-ray crystal structure (Fig. 5), the complex cation [1•La]$^{5+}$ has three acetate and two triflate counteranions, among which only the three acetate ions coordinated to the La$^{3+}$. This clearly demonstrates the complete binding selectivity of [1•La]$^{5+}$ for acetate ion over triflate ion. The higher coordination ability of the acetate ions to La$^{3+}$ probably contributes to the acceleration of the exchange of K$^+$ with La$^{3+}$ in the metallohost 1(OTf)$_2$ (Fig. 4).

## Discussion

We synthesized a dicationic cobalt(III) metallohost 1(OTf)$_2$ that has capping sites to hold the anion caps. The guest ion

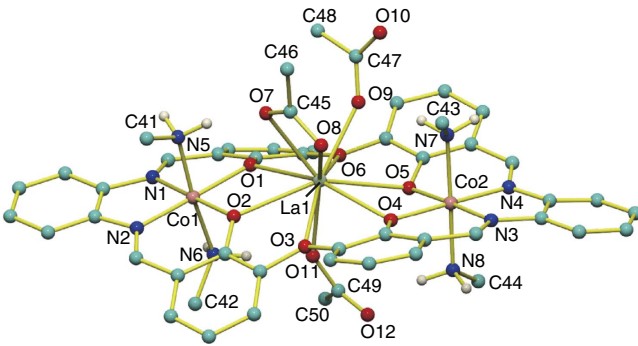

**Figure 5 | X-ray crystal structure of [1•La(OAc)₃](OTf)₂.** Hydrogen atoms are omitted for clarity except for those of the NH₂ groups. The solvent molecules and two triflate anions are also omitted.

($Na^+$, $K^+$, $Rb^+$, $Cs^+$, $Ca^{2+}$ or $La^{3+}$) was encapsulated in the $O_6$ cavity of $\mathbf{1}(OTf)_2$ in such a way that the two counteranions capped the guest cation. Uptake of $La(OTf)_3$ by $\mathbf{1}(OTf)_2$ was exceptionally slow (half-life ∼10 h) probably due to the capping effect of the triflate ion. This anion-capped structure enabled us to construct a host–guest system in which the guest exchange is accelerated in a time-programmable and on-demand manner. We believe that these results will open the way for the development of new functional materials in which the desired function is triggered by guest recognition not only by chemical stimuli but also by physical stimuli such as light or heat.

## Methods

**Materials and methods.** Reagents and solvents were purchased from commercial sources and used without further purification. $^1H$ NMR spectra were recorded on a JEOL JNM-ECS 400 (400 MHz). Chemical shifts in $CD_3OD$ were referenced with respect to the solvent residual peak (3.31 p.p.m.). Electrospray ionization time of flight (ESI-TOF) mass spectra were recorded on a Bruker Daltonics micrOTOF II spectrometer.

**Synthesis of [LCo₂(CH₃NH₂)₄](OTf)₂ (1(OTf)₂).** A solution of macrocyclic ligand $H_4L$ (149 mg, 0.201 mmol)[44,45] in chloroform (40 ml) was mixed with a solution of cobalt(II) acetate tetrahydrate (108 mg, 0.434 mmol) in methanol (12 ml), a solution of tetrabutylammonium trifluoromethanesulfonate (783 mg, 2.00 mmol) in methanol (12 ml) and then methylamine (40% in methanol, 2.0 ml). The resulting solution was stirred under air for 12 h. After the solution was concentrated to dryness, the residue was dissolved in acetonitrile and diethyl ether was added. The precipitated crude product was collected and dissolved in a small amount of methanol containing methylamine (40% in methanol, 1.0 ml). The solution was left for 3 h and then diethyl ether was added to give precipitates, which were collected by filtration. After repeating the purification by reprecipitation, the product was collected by filtration to give $\mathbf{1}(OTf)_2$ (150 mg, 0.125 mmol, 62%) as a reddish brown powder. $^1H$ NMR (400 MHz, CD₃OD) δ 8.80 (s, 4H), 8.34–8.29 (m, 4H), 7.61–7.56 (m, 4H), 7.51 (dd, $J = 7.8$, 1.4 Hz, 4H), 7.35 (dd, $J = 7.8$, 1.4 Hz, 4H), 6.72 (t, $J = 7.8$ Hz, 4H), 1.64 (s, 12H). ESI-MS $m/z$ 1047.1 $[\mathbf{1} + OTf]^+$. Anal. Calcd for $C_{46}H_{44}Co_2F_6N_8O_{12}S_2 \cdot 3MeOH \cdot 2H_2O$: C, 44.28; H, 4.55; N, 8.43. Found: C, 44.42; H, 4.81; N, 8.76.

**X-ray crystallography.** Intensity data were collected on a Rigaku Mercury diffractometer (with Mo Kα radiation, $\lambda = 0.71073$ Å) or a Bruker SMART APEX II diffractometer (with Cu Kα radiation, $\lambda = 1.54178$ Å). The data were corrected for Lorentz and polarization factors and for absorption by semi-empirical methods based on symmetry-equivalent and repeated reflections. The structures were solved by direct methods (SHELXS 97 (ref. 49) or SHELXT[50]) and refined by full-matrix least squares on $F^2$ using SHELXL 2014 (ref. 51) (Supplementary Tables 1-3 and Supplementary Figs 15-19).

**Data availability.** Crystallographic data in this paper can be obtained free of charge via www.ccdc.cam.ac.uk/data_request/cif (or from the Cambridge Crystallographic Data Centre, 12 Union Road, Cambridge CB2 1EZ, UK). The deposit numbers are 1525963 ([1•K(OTf)₂](OTf)), 1525964 ([1•La(OAc)₃](OTf)₂), 1525965 ([1•Na(OTf)₂](OTf)), 1525966 ([1•Ca(OTf)₂](OTf)₂) and 1525967

($\mathbf{1}(OTf)_2$). All other data are available in this article and in the Supplementary Information file.

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

## Acknowledgements

This work was supported in part by JSPS KAKENHI (Grant Number JP16H06510 (Coordination Asymmetry) and JP26288022), the Noguchi Institute and Kanazawa University CHOZEN Project. We thank Dr Kenji Yoza (Bruker AXS KK) for the X-ray data collection for [1●K(OTf)₂](OTf), [1●Ca(OTf)₂](OTf)₂ and [1●La(OAc)₃](OTf)₂.

## Author contributions

S.A. initiated the project and Y.S. designed the research; C.M. carried out all the experiments under the guidance of Y.S. and S.A. All authors contributed to analysing and interpreting the data, and Y.S. and S.A. co-wrote the manuscript.

## Additional information

**Competing interests:** The authors declare no competing financial interests.

