## [Peer Review File · Nature Communications]

Reviewers' comments:

Reviewer #1 (Remarks to the Author):

The paper reported by Akine et al described an interesting switching of molecular recognition selectivity based on Host-Guest chemistry of carefully designed host molecule. Upon changing the counterion that capped the cavity, the binding barrier was changed, which allows on-demand recognition of ions. The slow kinetics of host-guest recognition was employed to allow switching the kinetic trapped states transform into the thermal stable state. This elegant use of kinetic traps to realize functional control is a nice demo of learning from nature. Therefore it is a work deserves publication on Nature Communication.

However, about the recognition mechanism, the authors need to explain clearer. They claim the size matching between the metal ions and the host is crucial for the recognition, but the data in Table 1 are very confusing, since the size of metal ions didn't show proportional relation to the binding constant. Obviously, there are other factors than size matching, or there are errors in determination of the binding constants. If so, the authors should validate the method or provide the errors.

Reviewer #2 (Remarks to the Author):

The paper reports 5 related crystal structures, the parent 1(OTf) and 4 containing cationic guests; Na, K, Ca and La. The parent structure is well determined with only minor disorder in the methylamine. However, importantly for the discussion, as commented on by the authors the OTf anions do not occupy the capping sites in the 'empty' structure. This would appear to undermine the argument that these capping anions provide a removable barrier to the cation guest inclusion - at least from the crystal structure - as they only appear to occupy this position when there is a cation guest present.

The authors have put in a great deal of work to model the disorder in several of the structures and it appears to be very carefully done. However it may be attempting to model something which is not due to disorder - e.g. a larger cell or lower symmetry. No mention of this is given in the CIF or in the supplementary data about this and is needed to allow review of these structures.

The [1.Na] structure appears correct and the alert for H14 and H33 is probably due to an incorrectly placed H on the OH of the MeOH. It would be better to omit the H from the model (it's only 15% occupied MeOH) rather than incorrectly place it but ensure that it is included in the unit cell contents/formula.

[1.K] has very extensive disorder modelling and a great deal of careful work, however there is again no description provided. Given the extent of the disorder it would be helpful if the authors also gave details of any other attempts to resolve this - e.g. refining in P1, examination of data that a larger cell had not been missed.

[1.Ca] The level of disorder in this structure is very high. Disorder in all methylamine, Ca and coordinated OTf over 3 positions and while carefully modelled it could possibly be due to something else, for example an incorrect space group assignment. The Ca occupancy appears to be correct in the CIF for its symmetry site and disorder but it is unusual for checkcif to make this type of error. I am unable to determine if this structure is correct or not without further information. No res or hkl is provided within the CIF and no description of the model or other approaches to model the structure is provided and without this then I would not accept it for publication.

[1.La] Again a great deal of work has gone into the disorder model, however I don't think it is correct. The occupancy of the minor component modelled as 0.7 (OTf) over 2 sites is too low with very large - nearly 3 electron peaks - coinciding with O20. The 0.1 Cl is not convincing. There are

other residual peaks remaining in locations that are not chemically reasonable and may be due to twinning not completely accounted for. This refinement needs further work and description to be provided before it is publishable.

Reviewer #3 (Remarks to the Author):

The report by Akine and co-workers describes a dicobalt metallomacrocycle, which complexes cations within a crown-ether like pocket. Complexation is kinetically slow when the counterion to the metallomacrocycle is triflate, which the authors attribute to favourable hydrogen bonding between N-H groups from an axially coordinated methylamine co-ligand. Addition of acetate dramatically enhances the rate of cation binding to the central macrocycle. The compounds are well-characterised and the NMR experiments are compelling. The crystallography appears very well done, and the authors responses to checkCIF alerts are valid.

The paper has some novelty, and I suggest publication after some additional studies.

Suggested additional work/changes:

- In my opinion, Table 1 would benefit dramatically if the association constants were compared with known crown ethers or benzocrownethers. There is a wealth of information regarding these values in the literature (including in the same solvent), so this should not require too much additional work.
- A significant factor behind the slow binding is believed to be steric demands, and the shielding of the binding pocket by a H-bonded triflate anion. What effect does the co-ligand have on this? E.g. does something like t-butylamine lead to slow binding (with or without triflate anions)?
- It is implied (p. 7), but not explicitly stated that if the authors start from the F-/Cl-/Br-/BF₄-/PF₆-/BPh₄- salts of the metalocycle, cation binding is rapid. This should be clarified. Presumably these anions do not speed up binding of cations to 1.(OTf)₂ because they cannot displace the favourably H-bonded triflate anions. I think looking at the kinetics of cation binding to 1.(BPh₄)₂ would be a good "control" experiment, as this would measure "pure" cation binding without an effective cap.
- Presumably, the authors could start with a non-coordinating anion (i.e. something like 1.(BPh₄)₂) and then determine association constants for the anion binding of both triflate and acetate. I think this would be interesting (and helpful) - presumably acetate binds more strongly, but is smaller, and thus a less effective cap?
- I find Figure 1 incredibly confusing. I would suggest starting again from scratch, and getting rid of the (in my opinion) unhelpful reaction coordinate graphs.
- In Figure 2, I don't really find the X-ray crystal structure that easy to visualise. I would suggest showing this as a line/ball-and-stick drawing rather than an ellipsoid plot, and including the N-H protons.
- Are the authors sure about the association constant of 2400 for Cs⁺? At a concentration of 1 mM, a K_a of 2400 would imply very close to half of the cation being bound after one equivalent of Cs⁺. However, the NMR spectrum (Supp. Figure 7) shows only a trace of bound species.
- The final sentence of the introduction is "We achieved the first example of on-demand acceleration of the guest exchange from a kinetically-trapped state, which is triggered by the anion cap exchange." This is a relatively vague and ill-defined statement. I don't think the claim of primacy is necessary, and would rather just see a sentence summing up what the paper achieves here.

- On p.7, the authors state that guest binding to 1(OTf)₂ is much faster than crowns, and that "obviously the introduction of anion caps can quite effectively retard the guest inclusion." This is definitely a factor at play here, but this is perhaps not an entirely fair comparison, as crown ethers are quite different from the central ether-pocket of this 2+ macrocycles. I.e. the anion caps are a factor, but not the only one, and a less broad statement here would be more appropriate in my opinion.

- There are some minor typos/writing errors, which should be fixed before publication. E.g. "guest exchange to the more stable was significantly accelerated..." (abstract), "One of the examples is the passive transport across membranes... (p. 3)"

Response to the comments by the reviewers (NCOMMS-17-02954)

Response to Reviewer #1

Comment:

The paper reported by Akine et al described an interesting switching of molecular recognition selectivity based on Host-Guest chemistry of carefully designed host molecule. Upon changing the counterion that capped the cavity, the binding barrier was changed, which allows on-demand recognition of ions. The slow kinetics of host-guest recognition was employed to allow switching the kinetic trapped states transform into the thermal stable state. This elegant use of kinetic traps to realize functional control is a nice demo of learning from nature. Therefore it is a work deserves publication on Nature Communication.

However, about the recognition mechanism, the authors need to explain clearer. They claim the size matching between the metal ions and the host is crucial for the recognition, but the data in Table 1 are very confusing, since the size of metal ions didn't show proportional relation to the binding constant. Obviously, there are other factors than size matching, or there are errors in determination of the binding constants. If so, the authors should validate the method or provide the errors.

Response:

As we described in the original manuscript, the binding selectivity is mostly governed by the ionic radius of the guest ions based on the size-fit principle. However, as the reviewer pointed out, the binding constants are also influenced by other factors, such as charges of the guest ions. Therefore, we added a phrase "although the binding strengths might be influenced by some other factors such as the charge of the guest ions" in the revised manuscript. In addition to the above changes, we added information of the competitive experiments that were used for determination of binding constants for K^+ , Na^+ , Ca^{2+} , and La^{3+} (Supplementary Figure 11).

Response to Reviewer #2

Comment:

The paper reports 5 related crystal structures, the parent 1(OTf) and 4 containing cationic guests; Na, K, Ca and La. The parent structure is well determined with only minor disorder in the methylamine. However, importantly for the discussion, as commented on by the authors the OTf anions do not occupy the capping sites in the 'empty' structure. This would appear to undermine the argument that these capping anions provide a removable barrier to the cation guest inclusion - at

least from the crystal structure - as they only appear to occupy this position when there is a cation guest present.

Response:

As the reviewer pointed out, the capping sites are vacant when the guest is absent. This may not be a structure we had expected, but the guest uptake/release kinetics of $\mathbf{1}(\text{TfO})_2$ was significantly slow as compared with usual crown ethers. This means that the counteranions are involved in the guest uptake/release processes; extra energy is required to remove the counteranions at the capping sites in the transition state. This should be a reason why the “capping effect” is observed even though the empty structure is not “capped”. One of the significant achievements in this paper is the on-demand acceleration of guest exchange. In this case, the anion caps would retard both of the guest release and uptake processes, resulting in the unprecedented slow kinetics enough to give a very stable kinetically-trapped state. Therefore, we convince that the “uncapped” empty structure does not diminish the importance of our results.

Comment:

The authors have put in a great deal of work to model the disorder in several of the structures and it appears to be very carefully done. However it may be attempting to model something which is not due to disorder - e.g. a larger cell or lower symmetry. No mention of this is given in the CIF or in the supplementary data about this and is needed to allow review of these structures.

Response:

We have re-examined the structural analysis according to the comments by the reviewer and revised the analysis of some of the structures. We also added brief description about the refinement in the CIF file.

Comment:

The [1.Na] structure appears correct and the alert for H14 and H33 is probably due to an incorrectly placed H on the OH of the MeOH. It would be better to omit the H from the model (it's only 15% occupied MeOH) rather than incorrectly place it but ensure that it is included in the unit cell contents/formula.

Response:

- According to the suggestion, we deleted the four hydrogen atoms (H33, H34, H35, and H36) on the methanol molecule (O11–C29) in the structure of [1.Na]. The revised CIF and the checkcif report were uploaded. In the revised CIF files, the description about the disorder was included in

“_refine_special_details”.

Comment:

[1.K] has very extensive disorder modelling and a great deal of careful work, however there is again no description provided. Given the extent of the disorder it would be helpful if the authors also gave details of any other attempts to resolve this - e.g. refining in $P1$, examination of data that a larger cell had not been missed.

Response:

- According to the suggestion, we re-examined the structure determination of [1.K]. The refinement in the space group $P1$ gave a structure in which the positional disorder of K^+ and CF_3SO_3 anions was resolved, while the residuals ($R1$, $wR2$) and the coordination structure was not so different from those solved in the space group $P-1$. Since the Flack parameter in the $P1$ space group was refined to be nearly 0.5, the crystal was a racemic twin. The possibility for a larger-cell structure had been excluded by checking the diffraction image data. Therefore, the revised CIF and checkcif report for the structure in the space group $P1$ were uploaded. In addition, the description about structural analysis was included in “_refine_special_details”.

Comment:

[1.Ca] The level of disorder in this structure is very high. Disorder in all methylamine, Ca and coordinated OTf over 3 positions and while carefully modelled it could possibly be due to something else, for example an incorrect space group assignment. The Ca occupancy appears to be correct in the CIF for its symmetry site and disorder but it is unusual for checkcif to make this type of error. I am unable to determine if this structure is correct or not without further information. No res or hkl is provided within the CIF and no description of the model or other approaches to model the structure is provided and without this then I would not accept it for publication.

Response:

- As suggested by the reviewer, we investigated to solve the structure of [1.Ca] in the subgroups of $Ccca$ with lower symmetries, such as $C2cb(Aba2)$, Cc , $Ccc2$, $C222$, $C2/c$, $P2/c$, etc. However, the disorder was not resolved in any trials. Therefore, we concluded that the space group of the structure could be interpreted as $Ccca$. The details about the procedures and refinement are given in separate sheets (attached at the end of this document) and the brief summary of the refinement details was included in “_refine_special_details”. Since the refinement in the previous submission was carried out using SHELXL 97, we updated the analysis by using the new version SHELXL 2014, which gives the CIF file with the embedded res and hkl files.

Comment:

[1.La] Again a great deal of work has gone into the disorder model, however I don't think it is correct. The occupancy of the minor component modelled as 0.7 (OTf) over 2 sites is too low with very large - nearly 3 electron peaks - coinciding with O20. The 0.1 Cl is not convincing. There are other residual peaks remaining in locations that are not chemically reasonable and may be due to twinning not completely accounted for. This refinement needs further work and description to be provided before it is publishable.

Response:

- According to the reviewer suggestion, incorrectly assigned chlorine atoms were deleted and the sum of the occupancy factors of CF₃SO₃ anions was set to be 1.00. In addition, we re-investigated the refinement of solvent molecules (methanol molecule that is disordered over three positions and a water molecule) to give the residuals ($R1 = 0.1098$, $wR2 = 0.3140$), which were slightly larger than the initial refinement. The CIF and checkcif report for the revised structure were uploaded. In addition, the description about the structural analysis was included in “_refine_special_details”.

Response to Reviewer #3

Comment:

The report by Akine and co-workers describes a dicobalt metallomacrocyclic, which complexes cations within a crown-ether like pocket. Complexation is kinetically slow when the counterion to the metallomacrocyclic is triflate, which the authors attribute to favourable hydrogen bonding between N-H groups from an axially coordinated methylamine co-ligand. Addition of acetate dramatically enhances the rate of cation binding to the central macrocyclic. The compounds are well-characterised and the NMR experiments are compelling. The crystallography appears very well done, and the authors responses to checkCIF alerts are valid.

The paper has some novelty, and I suggest publication after some additional studies.

Suggested additional work/changes:

- In my opinion, Table 1 would benefit dramatically if the association constants were compared with known crown ethers or benzocrownethers. There is a wealth of information regarding these values in the literature (including in the same solvent), so this should not require too much additional work.

Response:

As the reviewer suggested, we included the association constants for 18-crown-6 with alkali metal cations determined under similar conditions (25 °C, methanol) in Table 1 and their references (refs 53 and 54).

53 Izatt. R. M., Pawlak. K., & Bradshaw. J. S. Thermodynamic and kinetic data for macrocycle interaction with cations and anions. *Chem. Rev.* **91**, 1721-2085 (1991)

54 Smetana, A. J. & Popov. A. I. Lithium-7 nuclear magnetic resonance and calorimetric study of lithium crown complexes in various solvents. *J. Solution Chem.* **9**, 183-196 (1980).

Comment:

- A significant factor behind the slow binding is believed to be steric demands, and the shielding of the binding pocket by a H-bonded triflate anion. What effect does the co-ligand have on this? E.g. does something like t-butylamine lead to slow binding (with or without triflate anions)?

Response:

We have also been interested in the effect of the structural differences of the co-ligand on the thermodynamic/kinetic parameters of the guest binding. We thus synthesized several analogous metallohosts that have secondary amines or pyridine as an axial ligand. However, they turned out to show almost no recognition ability towards guest cations such as Na⁺ and K⁺. Therefore, methylamine is essential for anchoring the triflate anion and for shielding of the binding pocket.

Comment:

- It is implied (p. 7), but not explicitly stated that if the authors start from the F-/Cl-/Br-/BF₄-/PF₆-/BPh₄- salts of the metallocycle, cation binding is rapid. This should be clarified. Presumably these anions do not speed up binding of cations to 1.(OTf)₂ because they cannot displace the favourably H-bonded triflate anions. I think looking at the kinetics of cation binding to 1.(BPh₄)₂ would be a good "control" experiment, as this would measure "pure" cation binding without an effective cap.

Response:

As the reviewer suggested, we added a brief description about the experiment in the main text as follows:

In order to investigate the effect of counteranions, 3 equiv of Bu₄NX (X = AcO⁻, F⁻, Cl⁻, Br⁻, BF₄⁻, PF₆⁻, BPh₄⁻) was added to the mixture of 1(OTf)₂ and La(OTf)₃ and the uptake rates were compared with the blank experiment. As we expected, AcO⁻ significantly accelerated the cation uptake, while

other anions (F^- , Cl^- , Br^- , BF_4^- , PF_6^- , BPh_4^-) did not. This acceleration was also observed when $La(OAc)_3$ was used as La^{3+} source.

As the reviewer suggested, we tried to synthesize $1(BPh_4)_2$ by a procedure similar to that for $1(OTf)_2$. Unfortunately, it was difficult to obtain pure sample of $1(BPh_4)_2$; we only obtained 1^{2+} having mixed counteranions, formulated approximately as $1(BPh_4)_{0.5}(X)_{1.5}$ ($X = OAc$ or OH). It was also difficult to remove the contaminated byproduct, Bu_4NBPh_4 . This is the reason why we used $1(OTf)_2$ as the starting complex for the evaluation of the effect of the counteranions.

Comment:

- Presumably, the authors could start with a non-coordinating anion (i.e. something like $1.(BPh_4)_2$) and then determine association constants for the anion binding of both triflate and acetate. I think this would be interesting (and helpful) - presumably acetate binds more strongly, but is smaller, and thus a less effective cap?

Response:

As described above, it was difficult to obtain pure sample of $1(BPh_4)_2$ that would be useful for directly evaluating the binding constants between the metallohost 1^{2+} and anions. However, as the reviewer pointed out, acetate ion binds more strongly to the metallohost 1^{2+} than triflate ion, which was clearly confirmed by the crystal structure of $[1\cdot La]^{5+}$ (Fig. 5). The complex cation $[1\cdot La]^{5+}$ has three acetate and two triflate counteranions, among which only the acetate ions coordinated to the La^{3+} . This preference could be explained in terms of basicity ($pK_a = 4.76$ for acetic acid and -14.7 for triflic acid). The description about the selectivity was included in the main text as follows:

This clearly demonstrates the complete binding selectivity of $[1\cdot La]^{5+}$ for acetate ion over triflate ion.

We think that this strong binding of acetate ion to La^{3+} should be responsible for the acceleration of the uptake rate and the guest exchange. A possible explanation, which mostly agrees with the reviewer's opinion, is as follows:

- The process of La^{3+} -uptake requires relatively high activation energy due to strong electronic repulsion between dicationic metallohost 1^{2+} and trivalent La^{3+} . However, acetate ion strongly interacts with La^{3+} to give a tightly-bound ion pair such as $[La(OAc)]^{2+}$, which could behave as a monovalent cation with a smaller effective charge. This could lower the energy barrier for the uptake of $La(OAc)_3$.

Comment:

- I find Figure 1 incredibly confusing. I would suggest starting again from scratch, and getting rid of the (in my opinion) unhelpful reaction coordinate graphs.

Response:

According to the suggestion by the reviewer, we revised the Figure 1 by getting rid of the picture of energy landscape. Also, the layout of the figure components was changed.

Comment:

- In Figure 2, I don't really find the X-ray crystal structure that easy to visualise. I would suggest showing this as a line/ball-and-stick drawing rather than an ellipsoid plot, and including the N-H protons.

Response:

According to the suggestion by the reviewer, the thermal ellipsoid plots in Figure 2b and Figure 5 were replaced by ball-and-stick drawings. In addition, the N-H hydrogen atoms on the methylamine ligands are shown in the revised figures. Similarly, we included the N-H hydrogen atoms in the thermal ellipsoid plots in Supplementary Figures 16–19.

Comment:

- Are the authors sure about the association constant of 2400 for Cs⁺? At a concentration of 1 mM, a K_a of 2400 would imply very close to half of the cation being bound after one equivalent of Cs⁺. However, the NMR spectrum (Supp. Figure 7) shows only a trace of bound species.

Response:

The association constant of 2295 /M for Cs⁺ was obtained by non-linear curve fitting of ¹H NMR chemical shift changes, which resulted from the fast guest exchange between the Cs⁺-complex and the free host. From the simulation curve, we can estimate the mole fraction of Cs⁺-complex as 52% when 1 equiv of Cs⁺ is present, as reviewer pointed out. In order to clarify this, we added a figure of non-linear curve fitting in Supplementary Figure S7b.

Comment:

- The final sentence of the introduction is "We achieved the first example of on-demand acceleration of the guest exchange from a kinetically-trapped state, which is triggered by the anion cap exchange." This is a relatively vague and ill-defined statement. I don't think the claim of primacy is necessary, and would rather just see a sentence summing up what the paper achieves here.

Response:

As the reviewer suggested, we modified the sentence as follows;

This enables us to obtain a kinetically-trapped state, from which on-demand acceleration of the guest exchange is triggered by the anion cap exchange.

Comment:

- On p.7, the authors state that guest binding to $1(\text{OTf})_2$ is much faster than crowns, and that "obviously the introduction of anion caps can quite effectively retard the guest inclusion." This is definitely a factor at play here, but this is perhaps not an entirely fair comparison, as crown ethers are quite different from the central ether-pocket of this $2+$ macrocycles. I.e. the anion caps are a factor, but not the only one, and a less broad statement here would be more appropriate in my opinion.

Response:

We agree with the reviewer's opinion that it is not fair to compare the newly synthesized $1(\text{OTf})_2$ with crown ethers because of the difference of the charge and the chemical structures. Thus, we revised the description as follows;

Thus, the guest inclusion by $1(\text{OTf})_2$ was much slower than that by simple crown ethers (within milliseconds to seconds)⁴⁶⁻⁴⁸. The retarded guest inclusion should mainly be attributed to the effect of the anion caps, although there are differences in the charge and the host structures that could also affect the guest uptake rates.

Comment:

- There are some minor typos/writing errors, which should be fixed before publication. E.g. "guest exchange to the more stable was significantly accelerated..." (abstract), "One of the examples is the passive transport across membranes... (p. 3)"

Response:

As the reviewer suggested, we correct mistakes in wording.

(abstract):

The guest exchange to the more stable state from this kinetically-trapped state is significantly accelerated by exchange of TfO^- anion cap by AcO^- in an on-demand fashion.

(p. 3):

For example, passive transport finely controls ion uptake and release using ion channels and carriers according to their concentration gradients.

Details for crystal structure determination of [1•Ca(OTf)₂](OTf)₂

1. Reduced cell and Bravais lattice

Initial data reduction gave a reduced cell with the dimension of

$$a = 10.9181, b = 10.9189, c = 24.0423 \text{ \AA}, \alpha = 90, \beta = 90, \gamma = 106.1054 \text{ deg}$$

which could be transformed into a *C*-centered orthorhombic lattice with the dimension of

$$a = 13.126, b = 17.452, c = 24.042 \text{ \AA}$$

Therefore, the Laue symmetry of the intensities were checked for *mmm*, *2/m*, and *-1* groups. The R_{int} values (for reflections with $2\text{-theta} < 38.4 \text{ deg}$) for the Laue groups are as follows:

$$\text{mmm: } 0.0997$$

$$2/m \text{ (unique axis a): } 0.1037$$

$$2/m \text{ (unique axis b): } 0.1075$$

$$2/m \text{ (unique axis c): } 0.0885$$

$$-1: 0.074$$

Therefore, the crystal can have each of the Laue symmetries of *mmm*, *2/m*, and *-1*.

2. Extinction

For (0,k,l) reflections:

$$\text{average } F^2/\text{sig}F^2 \text{ (l = even)} = 39.80$$

$$\text{average } F^2/\text{sig}F^2 \text{ (l = odd)} = 0.21$$

For (h,0,l) reflections:

$$\text{average } F^2/\text{sig}F^2 \text{ (l = even)} = 25.85$$

$$\text{average } F^2/\text{sig}F^2 \text{ (l = odd)} = 0.47$$

For (h,k,0) reflections:

$$\text{average } F^2/\text{sig}F^2 \text{ (h = even, k = even)} = 43.16$$

$$\text{average } F^2/\text{sig}F^2 \text{ (h = odd, k = odd)} = 0.42$$

This intensity statistics well corresponds to the extinction for the space group *Ccca* (#68) with glide planes for all the three axes. Therefore, the space group was determined to be *Ccca* or its subgroups.

3. Structure determination

The structure was initially solved by SHELXS-97 in the space group *Ccca* and refined by using SHEXL-2014. The unit cell was found to contain four molecules of [1.Ca] ($Z = 4$). Since the space group *Ccca* has 16 asymmetric units per unit cell, each of the asymmetric unit contains 1/4 of the [1.Ca] molecule. In fact, the molecule lies on an intersection the three crystallographic 2-fold axes, e.g., at the point of (0, 1/4, 1/4). Therefore, the [1.Ca] molecule could have an apparent 222

point-group symmetry (D_2 symmetry in the Schoenflies notation).

However, the Ca atom was slightly displaced from the center of the molecule by 0.42 Å and disordered over two positions on the a axis (two-fold axis); the coordinate was refined to be (0.0323, 0.25, 0.25). Associated with this disorder, the coordinating CF_3SO_3 anions were disordered over four positions (each has occupancy factor of 0.25 due to the requirement of the 222 point-group symmetry). In addition, the structure contains coordinating methylamine molecule disordered over two positions (occupancy factors 0.75:0.25) and non-coordinating CF_3SO_3 anions disordered over two positions (0.5:0.5; on the 2-fold axis along the c axis).

Since the crystal was solved as a highly disordered structure, we investigated to resolve the disorder in the subgroups with lower symmetries. We investigated all the possible subgroups, but since the important focus is whether or not the positional disorder of the Ca atom could be resolved, we first investigated the structure refinement in the space groups that do not have 2-fold axes along b and c axes.

- $C2cb$ ($Aba2$ after axis conversion):

The refinement gave two Ca atoms on the a axis (two-fold axis), which are separated by 0.83 Å and have almost equal occupancy factors. Therefore we concluded that refinement in this space group cannot resolve the disorder.

- Cc :

The refinement gave two Ca atoms, which are separated by 0.83 Å (almost the direction of the a axis) and have almost equal occupancy factors. Therefore we concluded that refinement in this space group can not resolve the disorder.

- Other groups:

In the refinement in the $Ccc2$, $C222$, $C2/c$, and $P2/c$ groups, the disorder of the Ca atom can not be resolved due to the crystallographic symmetry requirements. In the refinement in the $C2$, Pc , $P2$, $P1$, and $P-1$ groups, the least-squares calculations were unstable and the coordinates were not reliably determined.

Reviewers' comments:

Reviewer #1 (Remarks to the Author):

I am basically satisfied with the authors respond to my concern. So I think this article is ready for publication.

Reviewer #2 (Remarks to the Author):

The authors have addressed many of the queries and concerns raised about the crystallography reported. However, the structure of [1.Ca] needs to be re-examined. There are a number of alerts in the checkcif report, whilst only C-alerts, that should not have been ignored. The reported formula does not match the contents of the model and for C, S and Ca - this should only be the case for H atoms which have not been located/included in the model. The count from checkcif appears to match the model and the formula reported by the authors is in error. Some of the coordinates and adps for Ca1, S3 and C15 have been manually fixed when if this is correct due to lying on special positions (2-fold axis) then the refinement will automatically generate these constraints, which it doesn't. There is something odd about this model/refinement and the authors need to investigate and correct this and ensure that the formula and values determined from it are correct.

Reviewer #3 (Remarks to the Author):

The authors have carefully and thoroughly addressed all of my queries/comments/suggestions. I recommend this manuscript be accepted for publication.

My only minor suggestion is that the authors' description of why they did not start binding studies from 1(BPh4)2 (because of the difficulties in isolating this compound) be included as a footnote - as to me the question of why binding studies weren't started from 1(BPh4)2 is quite an obvious one.

Response to the comments by the reviewers (NCOMMS-17-02954A)

Response to Reviewer #2

Comment:

The authors have addressed many of the queries and concerns raised about the crystallography reported. However, the structure of [1.Ca] needs to be re-examined. There are a number of alerts in the checkcif report, whilst only C-alerts, that should not have been ignored. The reported formula does not match the contents of the model and for C, S and Ca - this should only be the case for H atoms which have not been located/included in the model. The count from checkcif appears to match the model and the formula reported by the authors is in error.

Response:

According to the comments, we checked the formula of "1·Ca" (= $\text{LCo}_2\text{Ca}(\text{CH}_3\text{NH}_2)_4(\text{CF}_3\text{SO}_3)_4$ where $\text{L}^{4-} = [\text{C}_{40}\text{H}_{24}\text{N}_4\text{O}_6]^{4-}$) and confirmed that the formula $\text{C}_{48}\text{H}_{44}\text{CaCo}_2\text{F}_{12}\text{N}_8\text{O}_{18}\text{S}_4$ that we had reported in the previous submission was correct. The discrepancy of the experimental and calculated formulas in the previous checkcif report is probably due to the problem of the checkcif function, when coordinates of some atoms in "negative" PART are manually fixed in the SHELXL (by adding 10 in the fractional coordinates). Actually, the problem has been fixed after removing the manual constraints for Ca1, S3, and C15 as stated below and the checkcif function has output the correct formula for the revised structure.

Comment:

Some of the coordinates and adps for Ca1, S3 and C15 have been manually fixed when if this is correct due to lying on special positions (2-fold axis) then the refinement will automatically generate these constraints, which it doesn't. There is something odd about this model/refinement and the authors need to investigate and correct this and ensure that the formula and values determined from it are correct.

Response:

In the previous refinements, we believed that Ca1, S3, and C15 should be located on the special positions (2-fold axis), because they were initially found at the positions on the 2-fold axis. However, the reviewer comments reminded us the other possibility; we refined the structure without the constraints to obtain a reasonable model in which Ca1, S3, and C15 are displaced from the 2-fold axis and no constraints are required for their coordinates and ADPs. Now the problems regarding the checkcif alerts pointed out by the reviewer have been fixed and the revised structure was deposited in the CCDC database.

Response to Reviewer #3

Comment:

The authors have carefully and thoroughly addressed all of my queries/comments/suggestions. I recommend this manuscript be accepted for publication.

My only minor suggestion is that the authors' description of why they did not start binding studies from $1(\text{BPh}_4)_2$ (because of the difficulties in isolating this compound) be included as a footnote - as to me the question of why binding studies weren't started from $1(\text{BPh}_4)_2$ is quite an obvious one.

Response:

As the reviewer suggested, we included the description about why we did not start the binding studies from $1(\text{BPh}_4)_2$. Since the editorial office advised us not to use footnotes but to include the information in the main text, we added the following sentences in the main text:

In order to clarify the effect of the counteranions, we carried out anion-exchange experiments using a metallohost with a weakly coordinating anion. Actually, non-coordinating anion such as BPh_4^- would be suitable for the starting anion, but it was difficult to obtain pure sample of $1(\text{BPh}_4)_2$. Thus, we chose weakly coordinating TfO^- anion as the starting anion; we added 3 equiv of Bu_4NX ($\text{X} = \text{AcO}^-, \text{F}^-, \text{Cl}^-, \text{Br}^-, \text{BF}_4^-, \text{PF}_6^-, \text{BPh}_4^-$) to the mixture of $1(\text{OTf})_2$ and $\text{La}(\text{OTf})_3$ and compared the uptake rates with the blank experiment.

REVIEWERS' COMMENTS:

Reviewer #2 (Remarks to the Author):

The authors have dealt with the specific issues raised for the 1.Ca structure. The model is possibly still more complicated than it needs to be, perhaps only needing the Ca to be shifted off the intersection of the two 2-fold axes but still on one of the 2-fold axes but the structure as submitted now is accepted.

Response to the comments by the reviewers (NCOMMS-17-02954B)

Response to Reviewer #2

Comment:

The authors have dealt with the specific issues raised for the 1.Ca structure. The model is possibly still more complicated than it needs to be, perhaps only needing the Ca to be shifted off the intersection of the two 2-fold axes but still on one of the 2-fold axes but the structure as submitted now is accepted.

Response:

As the reviewer suggested, we had carefully investigated whether Ca^{2+} is located on the 2-fold axes or not. We employed the current model with the displaced Ca^{2+} , because the coordinating TfO^- anions that are disordered over four positions do not have 2-fold symmetry.